# Interhemispheric Functional Connectivity in the Primary Motor Cortex Assessed by Resting-State Functional Magnetic Resonance Imaging Aids Long-Term Recovery Prediction among Subacute Stroke Patients with Severe Hand Weakness

**DOI:** 10.3390/jcm9040975

**Published:** 2020-04-01

**Authors:** Yu-Sun Min, Jang Woo Park, Eunhee Park, Ae-Ryoung Kim, Hyunsil Cha, Dae-Won Gwak, Seung-Hwan Jung, Yongmin Chang, Tae-Du Jung

**Affiliations:** 1Department of Rehabilitation Medicine, School of Medicine, Kyungpook National University, Daegu 41944, Korea; ssuni119@naver.com (Y.-S.M.); ehmdpark@naver.com (E.P.); ryoung20@hanmail.net (A.-R.K.); 2Department of Rehabilitation Medicine, Kyungpook National University Hospital, Daegu 41944, Korea; eodnjs108@naver.com (D.-W.G.); pyromyth@naver.com (S.-H.J.); 3Department of Biomedical Engineering, Seoul National University College of Medicine, Seoul 03080, Korea; 4Department of Medical & Biological Engineering, Kyungpook National University, Daegu 41944, Korea; giantstar.jw@gmail.com (J.W.P.); hscha1002@daum.net (H.C.); 5Department of Radiology, Kyungpook National University Hospital, Daegu 41944, Korea; 6Department of Molecular Medicine, School of Medicine, Kyungpook National University, Daegu 41944, Korea

**Keywords:** functional magnetic resonance imaging, neuronal plasticity, recovery of function, stroke, motor cortex

## Abstract

This study aimed to evaluate the usefulness of interhemispheric functional connectivity (FC) as a predictor of motor recovery in severe hand impairment and to determine the cutoff FC level as a clinically useful parameter. Patients with stroke (*n* = 22; age, 59.9 ± 13.7 years) who presented with unilateral severe upper-limb paresis and were confirmed to elicit no motor-evoked potential responses were selected. FC was measured using resting-state functional magnetic resonance imaging (rsfMRI) scans at 1 month from stroke onset. The good recovery group showed a higher FC value than the poor recovery group (*p* = 0.034). In contrast, there was no statistical difference in FC value between the good recovery and healthy control groups (*p* = 0.182). Additionally, the healthy control group showed a higher FC value than that shown by the poor recovery group (*p* = 0.0002). Good and poor recovery were determined based on Brunnstrom stage of upper-limb function at 6 months as the standard, and receiver operating characteristic curve indicated that a cutoff score of 0.013 had the greatest prognostic ability. In conclusion, interhemispheric FC measurement using rsfMRI scans may provide useful clinical information for predicting hand motor recovery during stroke rehabilitation.

## 1. Introduction

Stroke is the leading cause of adult disability worldwide, accounting for a majority of patients with upper-limb impairment. The degrees of spontaneous improvement vary according to the severity of upper-limb paresis. In patients with mild-to-moderate upper-limb paresis, spontaneous recovery, as reflected by improvements in clinical parameters including Fugl-Meyer assessment of the upper extremity (FMA-UE) scores, is mainly restricted to the first 4 weeks post-stroke [1].There is evidence in the literature that in stroke patients with mild-to-moderate impairment, the degree of initial deficits predicts outcome. In contrast to mildly impaired patients, it is relatively difficult to predict the spontaneous recovery pattern of upper-limb motor function in severely impaired patients. Clinical data alone cannot accurately predict arm recovery, particularly in patients with initial severe upper-limb impairment [2,3,4]. High inter-individual variability associated with recovery makes it difficult to predict arm recovery. However, very few severely impaired patients show late-onset motor recovery of the upper-limb [5]. Therefore, a prognostic biomarker reflecting functional long-term motor recovery is urgently required to decide the manner in which rehabilitation treatment strategies, including goal setting and effective treatment duration, for upper-limb recovery in severe hemiplegic stroke patients can be modified.

Recently, we reported that initial power spectral density (PSD) analysis of resting-state functional magnetic resonance imaging (rsfMRI) data can provide a sensitive prognostic predictor for patients with subacute stroke combined with severe hand disability [6]. PSD is measured as resting-state intrinsic neuronal activity in the frequency domain. In contrast to PSD, functional connectivity (FC) analysis is another approach to measure the resting-state intrinsic neuronal activity in the time domain using rsfMRI. Changes in FC value in the interhemispheric motor cortex (M1) after stroke are reportedly reflective of long-term recovery, and patients with good functional outcomes have greater FC values than patients with poor outcomes [7,8,9]. However, a recent study reported that differences in FC value in the interhemispheric M1 did not change over time with recovery [10]. Therefore, whether motor recovery after stroke can be predicted by the change in interhemispheric FC still remains controversial.

This study aimed to evaluate whether interhemispheric FC is useful for predicting upper-limb motor recovery among patients with severe hand impairment for whom it was difficult to predict the recovery pattern based on an initial clinical parameter. Therefore, addition of FC as a prognostic parameter for patients with severe hand deficits may eventually be useful for setting individualized therapeutic goals and strategies as well as for selecting patients for future trials.

## 2. Materials and Methods

### 2.1. Subjects

Twenty-two patients (59.9 ± 13.7 years; 9 males and 13 females) and 12 healthy subjects (60.2 ± 6.8 years; 8 males, 4 females) were included in this study. They were all right-handed. The inclusion criteria for patients were as follows: (1) unilateral ischemic stroke in the middle cerebral artery (MCA) territory confirmed by MRI, (2) first stroke, (3) age over 20, (4) hemiplegic motor deficit less than Gr 1 by manual muscle test present at the time of admission (Table 1). Patients with unstable medical conditions and those lost to follow-up were excluded.

All patients underwent resting functional magnetic resonance imaging about 1 month (27.8 ± 8.4) from stroke onset. We used Brunnstrom stage (hand score) as a parameter to assess clinical outcome at 1 month and 6 months after stroke onset. We included the patients with Brunnstrom stage 1 (flaccidity or absence of an active finger movement) but without any motor-evoked potential (MEP) responses of the affected hand at 1 month after stroke. Patients with severely impaired cognitive function [Mini-Mental State Examination (MMSE) < 24], severe visual or perceptual impairment, previous musculoskeletal abnormality, or damaged upper-limbs were excluded. All patients received individual physiotherapy training as well as cognitive training every day. The physiotherapy treatments comprised 30-min sessions two times per day for five days a week and included walking and balance training as well as individual exercise. Informed consent was provided to all patients according to OO University Institutional Review Board (2012-05-023).

### 2.2. Motor Task Functional Magnetic Resonance Imaging

Region of interest (ROI) of M1 for each participant was defined using motor task functional magnetic resonance imaging (fMRI). Motor task fMRI alternatively comprised three active periods and three rest periods, and each period was 30-s long. A light touch on the leg or hand was used to give a start signal at the start point of each period. Participants performed the motor task twice with the right and left hands and repeated flexion–extension during scanning. If any of the participants could not move their hand, they received assistance to perform passive movement.

To perform motor task fMRI data acquisition, T2-weighted echo-planar imaging sequences were used with the following parameters: TE (echo time) = 40 ms, TR (repetition time) = 3000 ms, Flip Angle (FA) = 90°, FOV (field-of-view) = 21 cm, acquisition matrix = 64 × 64, 4-mm thickness with no gap, and total scan time = 4 min and 12 s, with four dummy scans.

### 2.3. Resting-State fMRI

All fMRI data were obtained on a Signa Exite 3.0-T scanner (GE Healthcare, Milwaukee, WI, USA). All applicants were instructed to lie down comfortably and close their eyes during MRI scanning, but not fall asleep. The rsfMRI data were obtained using T2-weighted echo-planar imaging sequences using the following parameters: TE = 40 ms, TR = 2000 ms, FA = 90°, FOV = 22 cm, acquisition matrix = 64 × 64, 4-mm thickness with no gap, and total scan time= 8 min and 12 s, with six dummy scans.

Three-dimensional-fast spoiled gradient echo sequence [repetition time (TR) = 7.8 ms; echo time (TE) = 3 ms; inversion time = 450 ms; flip angle = 20; matrix = 256 × 256; field-of-view (FOV) = 24 mm; 1.3 mm thickness] was used for the acquisition of T1-weighted high-resolution anatomical images.

### 2.4. fMRI Data Analysis

Image preprocessing and statistical analyses of fMRI data were conducted using the statistical parametric mapping software SPM12 (http://www.fil.ion.ucl.ac.uk/spm/), implemented in MATLAB (Mathworks, Inc., Sherborn, MA, USA). By slice-timing, realignment, co-registration, and normalization, functional images were preprocessed into the Montreal Neurological Institute (MNI) template based on a standard stereotaxic coordinate system and spatial smoothing with 8-mm full-width at half-maximum (FWHM) Gaussian kernel. FMRI data are superimposed onto MNI space. The seed MNI coordinates for the patients were summarized in Appendix A (Appendix A).

### 2.5. Rest State Functional Connectivity

The seed-based method was used to determine resting-state functional connectivity (rsFC). In brief, FC CONN15 toolbox (http://web.mit.edu/swg/software.htm) was used to show a strong temporal correlation between bilateral M1 and supplementary motor area (SMA). The contralesion and ipsilesion (namely M1 and SMA; spheres of 5-mm radius) were identified using MarsBar ROI tool (http://marsbar.sourceforge.net/) on MNI coordinates. Four ROI positions (spheres of 5 mm radius), namely contralesional M1, ipsilesional M1, contralesional SMA, and ipsilesional SMA, were selected based on individual motor task results. Noise, cerebrospinal fluid, white matter, and motion parameters were used to correct time fluctuations in blood-oxygen-level-dependent (BOLD) signals as nuisance covariates, and a band-pass filter (range, 0.008 Hz–0.09 Hz) was used. FC scores between pairs of ROIs on each subject were calculated using the FC SPM12 toolbox.

### 2.6. Lesion Volume Analysis

The lesion volume associated with hand motor function in the stroke area was calculated based on the overlapping area of the lesion mask between the T1-weighted images and the template of the corticospinal tract (CST). T1-weighted images were taken by preprocessing, which involves co-registration and normalization to a T1-weighted template using the SPM12 software package. A stroke physiatrist, who was blinded to the study, manually drew the lesions by using MRIcro (http://www.mccauslandcenter.sc.edu/crnl/mricro). The CST template was constructed using a previously reported method of probabilistic tractography [11,12]. Probabilistic tractography was conducted for 26 healthy controls to reconstruct CST. The seed, target, waypoint, and exclusion mask were drawn as follows. Individual seed masks for each hemisphere were placed in the hand knob area of M1 (MNI coordinates (37, −25, 62); (−37, −25, 62)), and each participant used an established semi-automated pipeline. The target masks were basis pontis. The waypoint masks included the posterior limb of the internal capsules and cerebral peduncles. For CST, a mask covering trajectories at the tegmentum pontis was added to the mid-sagittal and basal ganglia exclusion masks as an additional exclusion mask. A total of 50,000 streamlines were sent from M1 to the spinal target masks in the ventral medulla oblongata. Three different thresholds at 0.5%, 1%, and 2% were established for CST output distributions. The average of each tract was calculated for each of the three thresholds by summing all individual threshold- and subject-specific trajectories.

### 2.7. Statistical Analysis

To assess differences in FC scores between the three groups for each pair of ROIs, an ANOVA F-test was performed; subsequently, post-hoc two-sample t-tests were conducted for carrying out further comparisons. All statistical analyses were performed using the Statistical Package for the Social Sciences (SPSS, Chicago, IL, USA). A *p* value of <0.05 was considered to be statistically significant.

Receiver-operating characteristic (ROC) curve analysis was performed to determine the cutoff value for the prognostic model of upper-limb stroke recovery by using the difference in FC score between ipsilesional and contralesional M1 at 1 month.

True-positive rate (sensitivity) and false-positive rate (1-specificity) were computed and plotted as ROC curves. In an ROC space, a diagonal line corresponds to random discrimination. The area under the ROC curve (AUC) is commonly used to quantify classifier discriminability, with a value of 0.5 corresponding to random classification and a value of 1 corresponding to perfect classification.

## 3. Results

At 6 months after stroke onset, 11 patients (60.3 ± 15.9 years; seven males, four females) with Brunnstrom stage 4 (lateral prehension with release by thumb movement or semi-voluntary finger extension of a small range of motion) or 5 (palmar prehension or cylindrical/spherical grasp with limited function or voluntary mass finger extension of variable range) were categorized into the good recovery group and 11 patients (59.5 ± 12.9 years; six males, five females) with Brunnstrom stage 1, 2, and 3 were categorized into the poor recovery group. There were no age and sex-based differences between the good recovery and the poor recovery groups and the healthy control group (*p* = 0.986 and *p* = 0.827, respectively). Additionally, there was no statistical difference (*p* = 0.158) in lesion overlap volume between the good recovery group (0.33 ± 0.15 cc) and the poor recovery group (0.40 ± 0.17 cc). However, there was statistical difference (*p* = 0.019) in total lesion volume between the good recovery group (29.37 ± 41.31 cc) and the poor recovery group (125 ± 118.73 cc) (Figure 1). The demographic and clinical characteristics of 22 patients with stroke are summarized in Table 1.

Among the three groups, ANOVA F-test results reveal a statistically significant difference in FC between ipsilesional M1–contralesional M1 (*p* = 0.00039) (Figure 2). Post-hoc two-sample t-tests were performed for comparing the three groups further. The good recovery group showed a higher FC than that of the poor recovery group (*p* = 0.034). Contrastingly, the good recovery group showed no statistical difference in FC when compared with the healthy control group (*p* = 0.182), but the latter had a higher FC than that of the poor recovery group (*p* = 0.0002).

Moreover, according to the ANOVA F-test, the FC between ipsilesional SMA and contralesional SMA was significantly different among the three groups (*p* = 0.003) (Figure 1). In the post-hoc two-sample t-test, the FC between ipsilesional SMA and contralesional SMA was higher in the healthy control group compared with that in the good recovery group and the poor recovery group (*p* = 0.019 and *p* = 0.004, respectively), but there was no difference in the FC value between the good recovery group and the poor recovery group (*p* = 0.804).

Contrastingly, the ANOVA F-test result reveals no significant difference in FC among ipsilesional M1-SMA (*p* = 0.318), ipsilesional M1-contralesional SMA (*p* = 0.056), contralesional M1-ipsilesional SMA (*p* = 0.297), and contralesional M1-contralesional SMA (*p* = 0.656).

When the total lesion volume was included as a covariate in statistical analysis, ANOVA F-test results reveal a statistically significant difference in FC between ipsilesional M1–contralesional M1 among the three groups (*p* = 0.018) and in FC between ipsilesional M1–contralesional SMA among the three groups (*p* = 0.015). In the post-hoc two-sample t-test, however, there was no difference in the FC value between the good recovery group and poor recovery group (*p* = 0.232).

FC between ipsilesional and contralesional M1 positively correlated with hand function prognosis, as evaluated by Brunnstrom motor stages (BMS) (r = 0.581 and *p* = 0.005, respectively) (Figure 3). However, FC between ipsilesional and contralesional SMA did not correlate with hand function prognosis, as evaluated by BMS (r = −0.006, *p* = 0.979).

Good and poor recovery outcomes based on the Brunnstrom stage of upper-limb function at 6 months were determined as the standard, and the ROC curve indicated that a cutoff score of 0.013 had the greatest prognostic ability (maximum sensitivity and specificity) (Figure 4). The sensitivity of this model for predicting good recovery was 81.8% and the specificity was 63.6%. AUC value was 0.793, which is a fair level. 

## 4. Discussion

Here, we demonstrated the predictive value of FC for long-term motor outcomes in subacute stroke patients with stroke, predominantly in the MCA territory. The FC between ipsilateral and contralateral M1 was lower in the poor recovery group compared with that in the good recovery and healthy control groups, and it was also well correlated with hand function prognosis, evaluated by BMS. Our study demonstrated that the interhemispheric FC score calculated at 1 month after stroke has a good predictive value for recovery over a 6-month period in patients with severe hand impairment. However, the total lesion volume showed a tendency to swallow a large amount of explanatory variance of the outcome parameter. We also suggested an FC cutoff score for discriminating the good recovery group from the poor recovery group with high sensitivity and specificity.

Previous rsfMRI studies on acute stroke demonstrated that patients with mild-to-moderate motor deficits showed low interhemispheric FC score between the motor cortices [8,9,13]. These studies also revealed that the low FC score gradually increased during the recovery process and finally restored to near-normal levels in a good recovery group, while the FC score in the poor recovery group continued to decrease. Our findings are in line with these previous studies, i.e., interhemispheric FC score is useful for assessing the stroke prognosis and recovery despite varying scanning times (e.g., 3 days vs. 4 weeks) and evaluation domains [7,14,15].

These results suggest that functional neuroadaptation (reorganization) may be occurring in the most severely injured brains. One possible explanation for this finding is the physiological balance in reciprocal inhibitory projections between both of the hemispheres [16]. This explanation suggests that an abnormal inhibitory influence of the undamaged contralesional motor cortex on the damaged ipsilesional motor cortex disturbs the balance between the hemispheres, which is important for voluntarily generating paretic hand movement in poor recovery patients [16,17,18,19]. Here, the patients in the good recovery group had a higher rsFC between the motor cortex, which enhanced motor ability in the paretic hand. Our findings are consistent with those of previous studies, which demonstrated that rsFC within either the ipsilesional primary sensorimotor cortex or contralesional primary sensorimotor cortex reduced at an early stage after stroke, after which, in those patients who showed an improvement in motor impairment, the rsFC gradually increased to near-normal levels during recovery.

However, Nijboer et al. reported that ipsilesional rsFC between motor areas was lower than the contralesional rsFC, but this difference did not change over time [10]; they demonstrated that no relations were observed between individual changes in rsFC and upper-limb motor recovery. In that study, patient population presented with mild upper-limb impairments as opposed to the patients in our study who had severe motor impairment at 4 weeks after stroke. The patients only showed a limited amount of improvement (i.e., ceiling effect) after the first 4 weeks. The changes in brain activation patterns (i.e., cerebral reorganization) might have a different impact on a mild patient population compared with that on a population comprising severely impaired patients [20]. Here, patients were completely motor deficit with no hand movement and MEP response.

Using rsfMRI, we elucidated the cutoff value of FC to be 0.013, and sensitivity and specificity rates for good recovery prediction were 81.8% and 63.6%, respectively. There have been many studies on modeling the prediction of function recovery after stroke. The representative models are the PREP algorithm and the proportional recovery model [2,21]. However, when the two models were validated, the predicted prognosis rate (sensitivity and specificity) remained around 73–88%. The reason is that fitting to that predictive model did not work well in patients with severe corticospinal tract damage early in the injury, clinically complete unilateral paralysis, and patients with no MEP response. That is, in the case of mild to moderate severity, the prediction through clinical data and infarction size fit well, but in severe cases, it was difficult to predict through the model, and these were called ‘non-fitter’. In this study, we included relatively homogenous patients with MCA infarction, clinically no hand movement at all, and neurophysiologically no MEP response. Therefore, measuring interhemispheric functional connectivity in these patient populations can help predict prognosis. The prediction power associated with the use of only one parameter, namely interhemispheric rsFC, was comparable to the power associated with the multi-parameter model, without causing any compromise in sensitivity and specificity rates.

The present study had some limitations. First of all, our study is limited by small numbers in the patient population. Due to this limitation, stroke patients included only those with cortical and subcortical stroke, and we could not definitely determine the differential impact of lesion location and stroke severity on rsFC. Hence, from our results, we could not fully elucidate the mechanisms responsible for the reduction in rsFC after stroke as well as its influence on patient behavior. However, a more important clinical implication would be the establishment of the prognostic power of rsFC at an early stage. Therefore, larger sample size and longitudinal follow-up are warranted to confirm these relationships in future studies.

## 5. Conclusions

Interhemispheric FC estimated via rsfMRI provides useful clinical information and has a predictive value for hand motor recovery during stroke rehabilitation.

## Figures and Tables

**Figure 1 jcm-09-00975-f001:**
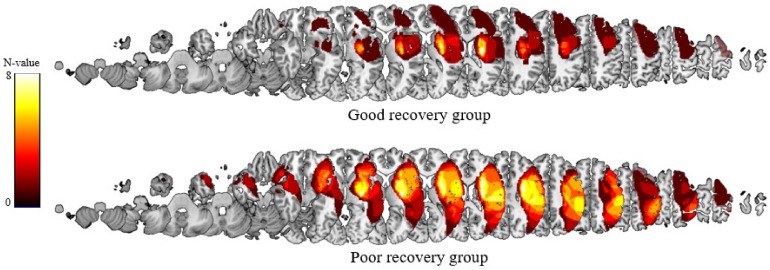
Total lesion overlay maps for the good recovery group and the poor recovery group.

**Figure 2 jcm-09-00975-f002:**
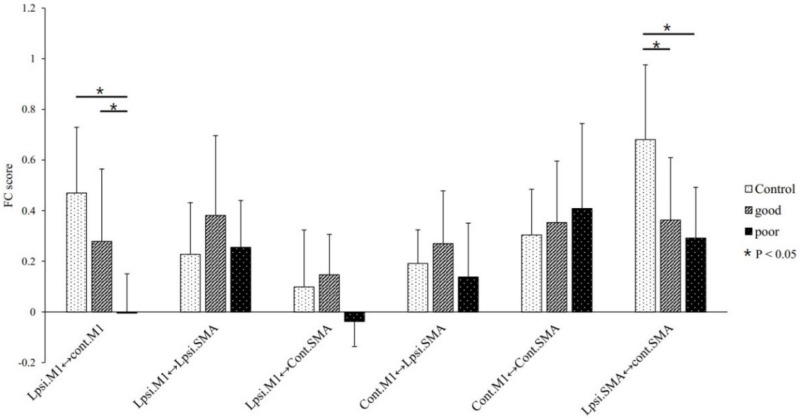
ANOVA F-tests showed significant differences in functional connectivity (FC) between ipsilesional M1-contralesional M1 among the three groups (*p* = 0.00039). Post-hoc two-sample t-tests were performed for further comparing between the groups. The good recovery group showed a higher FC than that shown by the poor recovery group (*p* = 0.034). In contrast, no significant difference in FC was seen between the good recovery and the healthy control groups (*p* = 0.182). Additionally, the healthy control group showed a higher FC than that of the poor recovery group (*p* = 0.0002).

**Figure 3 jcm-09-00975-f003:**
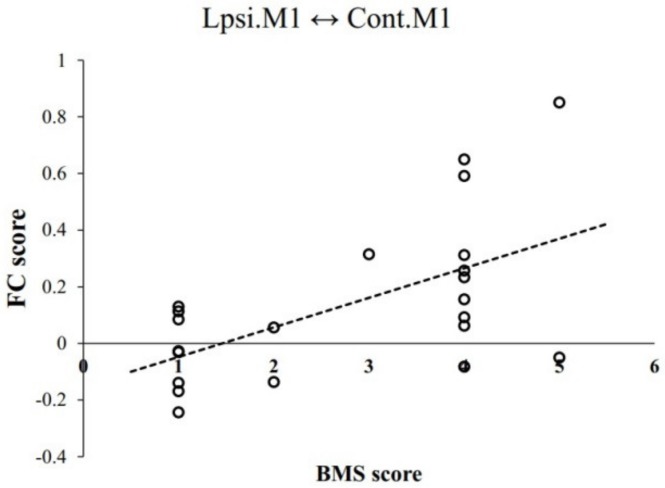
FC between ipsilesional and contralesional M1 is positively correlated with prognosis of hand function, as evaluated by Brunnstrom motor stages (BMS) (r = 0.581, *p* = 0.005).

**Figure 4 jcm-09-00975-f004:**
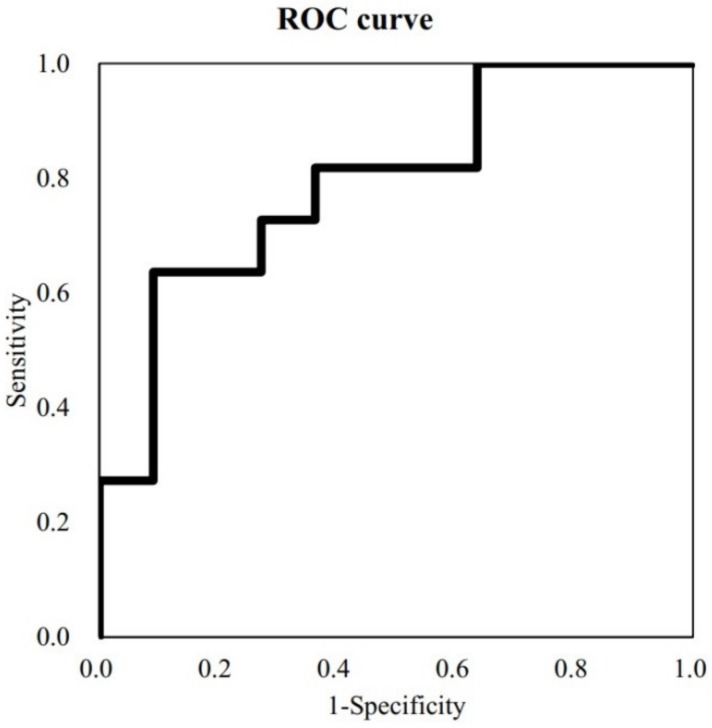
Good and poor recovery were determined based on Brunnstrom stage of upper-limb function at 6 months as the standard, and ROC (Receiver-operating characteristic) curve indicated that a cutoff score of 0.013 had the greatest prognostic ability (maximum sensitivity and specificity).

**Table 1 jcm-09-00975-t001:** Demographics and baseline characteristics of enrolled patients.

Subject	Group	Sex	Age	Lesion Territory	Total Lesion Volume (cc)	Lesion Volume (CST-Overlapped) (cc)	BS-Hand (Pre)	BS-Hand (Post)	Hand Dominance	BDI	MMSE	NIHSS
1	Good	F	57	MCA	4.8	0.247	1	4	Rt	10	28	8
2	Good	F	67	MCA	58.3	0.359	1	4	Rt	12	25	3
3	Good	F	70	MCA	13.1	0.439	1	4	Rt	22	27	9
4	Good	F	32	MCA	75	0.683	1	4	Rt	12	30	9
5	Good	F	80	MCA	4.7	0.226	1	4	Rt	10	27	5
6	Good	M	67	MCA	11.2	0.177	1	5	Rt	16	28	7
7	Good	F	75	MCA	7.1	0.241	1	4	Rt	24	26	9
8	Good	F	75	MCA	2.1	0.305	1	4	Rt	16	23	4
9	Good	M	40	MCA	9.0	0.216	1	4	Rt	23	27	9
10	Good	M	57	MCA	7.5	0.189	1	4	Rt	8	18	6
11	Good	M	44	MCA	130.4	0.544	1	5	Rt	5	14	7
12	Poor	F	66	MCA	84.5	0.522	1	1	Rt	33	5	16
13	Poor	M	42	MCA	273.8	0.246	1	1	Rt	20	24	7
14	Poor	M	59	MCA	268.7	0.680	1	2	Rt	4	24	9
15	Poor	F	75	MCA	78.3	0.291	1	1	Rt	29	-	13
16	Poor	F	75	MCA	25.0	0.257	1	1	Rt	13	25	13
17	Poor	M	68	MCA	23.6	0.247	1	1	Rt	5	21	15
18	Poor	M	40	MCA	334.2	0.442	1	1	Rt	-	-	21
19	Poor	F	69	MCA	121.4	0.683	1	1	Rt	15	24	14
20	Poor	F	44	MCA	164.4	0.571	1	1	Rt	-	-	12
21	Poor	M	53	MCA	5.1	0.302	1	2	Rt	2	30	6
22	Poor	F	64	MCA	5.3	0.245	1	3	Rt	28	30	11

CST; CorticoSpinal Tract, BS, Brunnstrom stage; BDI, Beck Depression Inventory; MMSE, Mini Mental State Examination; NIHSS, National Institutes of Health Stroke Scale MCA, Middle Cerebral Artery.

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
