# Peer review of "Interhemispheric Functional Connectivity in the Primary Motor Cortex Assessed by Resting-State Functional Magnetic Resonance Imaging Aids Long-Term Recovery Prediction among Subacute Stroke Patients with Severe Hand Weakness"

_jcm, 2020, doi:10.3390/jcm9040975_

Round 1

Reviewer 1 Report

The authors present an important topic under study evaluating the interhemispheric functional connectivity using functional MRI as a predictor of motor recovery in hand impairment. Their study while limited by small numbers is important to add to the literature on this topic. I suggest a few minor changes to this literature. 

1. Line 49 --it should be mild to moderate upper limb paresis not the other way round as has been written

2. Lines 175 -186 appear to be in error, they read as follows : Materials and Methods should be described with sufficient .....

3. The discussion gets too lengthy and the reader loses focus and track of the topic, would suggest making it a bit more concise if possible. 

4. The authors should acknowledge that their data is also limited by small numbers which is a limitation of the study. 

Author Response

Author responses to the reviewers

We thank the reviewer for his/her attention and thoughtful comments. Below we have provided responses to specific comments, and noted changes in the manuscript.

Reviewer 1

Comments and Suggestions for Authors

The authors present an important topic under study evaluating the interhemispheric functional connectivity using functional MRI as a predictor of motor recovery in hand impairment. Their study while limited by small numbers is important to add to the literature on this topic. I suggest a few minor changes to this literature. 

  1. Line 49 --it should be mild to moderate upper limb paresis not the other way round as has been written

(Author response) We thank the reviewer for kind correction. As commented by the reviewer, we corrected “moderate-to-mild” as “mild to moderate”.

  1. Lines 175 -186 appear to be in error, they read as follows : Materials and Methods should be described with sufficient .....

(Author response) We deleted line 175-186. This is where the journal's publication instruction was accidentally inserted. Thank you for your kind comment.

  1. The discussion gets too lengthy and the reader loses focus and track of the topic, would suggest making it a bit more concise if possible. 

(Author response) We thank the reviewer for thoughtful comment. Following the comment by the reviewer, we revised a very lengthy paragraph in Discussion session concisely. The deleted lengthy paragraph and the revised concise paragraph are as follows:

(The deleted paragraph) “Park et al. investigated rsfMRI in 12 subacute stroke patients and found a positive association of 6-month motor outcome and rsFC of the ipsilesional M1 with the contralesional thalamus, SMA, and medial frontal gyrus.[14] That study showed a longitudinal change in the interhemispheric connection during motor recovery; however, in our study, we cross-sectionally evaluated which change in FC has the potential to predict motor recovery prognosis at 4 weeks. The other study suggested that patterns in cortical connectivity might serve as a potential biomarker for the neural substratum associated with motor recovery after stroke in the chronic phase. Yin et al. revealed that the degree of connectivity in both the ipsilesional and contralesional M1 within the contralateral primary sensorimotor cortex was reduced in the completely paralyzed hand (CPH) group after subcortical stroke. The connectivity of these regions was positively correlated with Fugl–Meyer Assessment scores. Although these results were derived only from subcortical stroke patients, it is in agreement with our findings pertaining to the significance of interhemispheric FC in motor recovery among cortical stroke patients.[15] Puig et al. found that patients with good functional outcomes had greater FC score than those with poor functional outcome.[7]” “In such brains, interhemispheric connectivity is particularly reduced, which may cause compensatory increases in the same-side connectivity, involving cortico-subcortical connections (e.g., left caudate with left aITG) and typical crossed connections between the right cerebellum and the left cerebellar hemisphere. Our results are in line with the results of the aforementioned study, which revealed a reduction in interhemispheric FC after stroke in poor outcome patients, despite varying scanning times (3 days vs. 4 weeks) and evaluation domains. The aforementioned study used mRS as a general functional parameter, whereas we evaluated hand motor function recovery according to Brunnstrom stage.”

(The revised paragraph) These results suggest that functional neuroadaptation (reorganization) may be occurring in the most severely injured brains. One possible explanation for this finding is the physiological balance in reciprocal inhibitory projections between both the hemispheres.[16] This explanation suggests that an abnormal inhibitory influence of the undamaged contralesional motor cortex on the damaged ipsilesional motor cortex disturbs the balance between the hemispheres, which is important for voluntarily generating paretic hand movement in poor recovery patients.[16–19] Here, the patients in the good recovery group had a higher rsFC between the motor cortex, which enhanced motor ability in the paretic hand. Our finding was consistent with those of previous studies, which demonstrated that rsFC within either the ipsilesional primary sensorimotor cortex or contralesional primary sensorimotor cortex reduced at an early stage after stroke, after which in those patients who showed an improvement in motor impairment, the rsFC gradually increased to near-normal levels during recovery.

  1. The authors should acknowledge that their data is also limited by small numbers which is a limitation of the study. 

(Author response) We completely agree with the reviewer on that it should be acknowledged that our study is also limited by small numbers which is a limitation of the study. Therefore, we added it as study limitation as follows:

“The present study had some limitations. First of all, our study is limited by small numbers in patient population. Due to this limitation, stroke patients included only those with cortical and subcortical stroke, and we could not definitely determine the differential impact of lesion location and stroke severity on rsFC.”

Reviewer 2 Report

The authors conducted a clinical-imaging study on 22 stroke patients with 1 imaging data point (1 month) and two behavioral data points (1 and 6 months) after stroke. The research question whether or not the functional connectivity between motor regions measured in resting state fMRI can predict functional outcome of hand motor function is answered with Yes.

The paper is well written and comprehensive. However, here are some concerns or questions to the authors:

1) Is the MCA territory as infarction location, which you state in the "METHODS" section, an inclusion criterion or was it rather a RESULT of your given cohort? I'm missing the INclusion criteria, you give exclusion criteria, only. My main question: did you include based on a given (good or poor) motor deficit in the affected hand oder did you include based on a given stroke lesion location? Pontine infarctions might also result in hand motor deficit, why didn't you have any of those patients?

2) Deriving from the first question, authors should clearly differentiate, if given data are MATERIAL&METHODS or RESULTS. In the methods section, you write about the number of patients that you separated in the poor and in the good recovery group after 6 months. This might be a RESULT of your study and should therefore be placed in the results section of your paper. Please reconsider this paragraph.

3) The ROI definition for M1 and SMA based on task-fMRI seem to have major concerns, in my view. Because heavily affected patient simply cannot move their hand and you "assisted with passive movements" in those patients, this method might bring up very different areas including somatosensory areas and so on. As of now, I am not totally convinced in this method. Couldn't you simply rely on any given template atlas such as freesurfer and define M1 and SMA in your patients based on pre-existing atlases?

4) However, since this is an imaging paper, I would like to see an imaging figure, giving the following information: lesion locations (e.g. as summary lesion overlay map) and secondary, the seed MNI coordinates for the patients. The reader wants to be sure that your seed coordinates for the ROIs don't gap a priori between your groups.

5) Your "lesion volume" given in the table is actually not a total lesion volume but rather the overlap volume between the stroke lesion and the corticospinal tract. This makes totally sense to give and compare those numbers between the groups but I would expect to know the total lesion volume between the groups, too. Is there any statistical difference?

6) How do you interpret the slight however non-significant increased FC in contralesional M1 and SMA in poor recovery group? Is it kind of an compensatory upregulation?

7) Taken together, what exactly is the benefit of your proposed FC ROC cut-off value for prognosis in comparision with clinical measures and infarct location? As an outlook, authors may state a need of a future comparison study that directly compares the clinico-imaging prediction versus their given neuroscientific measures. From the clinical perspective, why should clinicians today get the suggested FC values for prognosis and not rely on the given clinical data plus the infarct site? Maybe the authors can speculate in the discussion a bit on that

8) 22 stroke patients doesn't seem that large population. Authors should consider to mention this in the limitations.

9) Line 299: this sentence is misunderstanding

10) Lines 175-186: somehow journal instructions on how to write the paper got into this section

Author Response

Author responses to the reviewers

We thank the reviewer for his/her attention and thoughtful comments. Below we have provided responses to specific comments, and noted changes in the manuscript.

Reviewer 2

Comments and Suggestions for Authors

The authors conducted a clinical-imaging study on 22 stroke patients with 1 imaging data point (1 month) and two behavioral data points (1 and 6 months) after stroke. The research question whether or not the functional connectivity between motor regions measured in resting state fMRI can predict functional outcome of hand motor function is answered with Yes.

The paper is well written and comprehensive. However, here are some concerns or questions to the authors:

1) Is the MCA territory as infarction location, which you state in the "METHODS" section, an inclusion criterion or was it rather a RESULT of your given cohort? I'm missing the INclusion criteria, you give exclusion criteria, only. My main question: did you include based on a given (good or poor) motor deficit in the affected hand oder did you include based on a given stroke lesion location? Pontine infarctions might also result in hand motor deficit, why didn't you have any of those patients?

(Author response) We thank the reviewer for thoughtful comment. Following the comment by the reviewer, we added a sentence to state our inclusion criteria clearly. Actually, the inclusion criteria include both lesion location in terms of MCA territory and motor deficit. The added sentence is as follows:

“The inclusion criteria for patients were as follows: (1) unilateral ischemic stroke in the middle cerebral artery (MCA) territory confirmed by MRI,(2) first stroke (3) age over 20, (4) hemiplegic motor deficit less than Gr 1 by manual muscle test present at the time of admission. Patients with unstable medical conditions and those lost to follow-up were excluded.”

2) Deriving from the first question, authors should clearly differentiate, if given data are MATERIAL&METHODS or RESULTS. In the methods section, you write about the number of patients that you separated in the poor and in the good recovery group after 6 months. This might be a RESULT of your study and should therefore be placed in the results section of your paper. Please reconsider this paragraph.

(Author response) Following the comment by the reviewer, we moved the paragraph from MATERIAL&METHODS to RESULTS.

3) The ROI definition for M1 and SMA based on task-fMRI seem to have major concerns, in my view. Because heavily affected patient simply cannot move their hand and you "assisted with passive movements" in those patients, this method might bring up very different areas including somatosensory areas and so on. As of now, I am not totally convinced in this method. Couldn't you simply rely on any given template atlas such as freesurfer and define M1 and SMA in your patients based on pre-existing atlases?

(Author response) We thank the reviewer for this very important and thoughtful comment. We completely agree with the reviewer in that the ROI definition for M1 and SMA based on task-fMRI might have a chance to bring up very different areas including somatosensory areas. However, we thought that the ROI definition for M1 and SMA based on any given template atlas might also have a chance to bring up very different M1 and SMA because these M1 and SMA based on any given template atlas are based on healthy subjects. The M1 and SMA of stroke patients might have a chance to relocate due to brain damage. This was a main reason why we used the ROI definition for M1 and SMA based on task-fMRI although there is a chance to bring up very different areas including somatosensory areas. Fortunately, the ROI definition for M1 and SMA were very similar between two methods. For example, in case of good recovery group, the right M1 was (x = 41.3, y= -22.4, z= 54.7) from task-fMRI and (x = 42, y= -22, z= 56) from the SPM template in MNI coordinates. The left M1 was (x = -38.5, y= -22.8, z= 54.4) from task-fMRI and (x = -38, y= -22, z= 56) from the SPM template in MNI coordinates. Other ROI definitions also showed the close similarity between two methods.

4) However, since this is an imaging paper, I would like to see an imaging figure, giving the following information: lesion locations (e.g. as summary lesion overlay map) and secondary, the seed MNI coordinates for the patients. The reader wants to be sure that your seed coordinates for the ROIs don't gap a priori between your groups.

(Author response) We added a figure which show the total lesion overlay maps in patient groups as Figure 1. The seed MNI coordinates for the patients were provided as a supplemental table S1 in Supplemental Information.

5) Your "lesion volume" given in the table is actually not a total lesion volume but rather the overlap volume between the stroke lesion and the corticospinal tract. This makes totally sense to give and compare those numbers between the groups but I would expect to know the total lesion volume between the groups, too. Is there any statistical difference?

(Author response) As expected by the reviewer, there was statistical difference in the total lesion volumes between the groups (P=0.019). Poor recovery group showed larger total lesion volume (125 ±118.73cc) than that (29.37±41.31cc) of good recovery group. We added a sentence regarding the total lesion volume between the groups in Results session as follows:

“However, there was statistical difference (p=0.019) in total lesion volume between the good recovery group (29.37±41.31cc) and poor recovery group (125 ±118.73cc).”

6) How do you interpret the slight however non-significant increased FC in contralesional M1 and SMA in poor recovery group? Is it kind of an compensatory upregulation?

(Author response) We thank the reviewer for very thoughtful comment. As comment by the reviewer, the slight however non-significant increased FC in contralesional M1 and SMA in poor recovery group might be a compensatory upregulation. However, we did not interpret it because we are careful for over-interpretation with two reasons. First, the increase in FC is statistically non-significant. Second, the increased FC in contralesional M1 and SMA in poor recovery group was not associated with clinically meaningful outcome.

7) Taken together, what exactly is the benefit of your proposed FC ROC cut-off value for prognosis in comparision with clinical measures and infarct location? As an outlook, authors may state a need of a future comparison study that directly compares the clinico-imaging prediction versus their given neuroscientific measures. From the clinical perspective, why should clinicians today get the suggested FC values for prognosis and not rely on the given clinical data plus the infarct site? Maybe the authors can speculate in the discussion a bit on that

(Author response) We thank the reviewer for this thoughtful and important comment. We revised the paragraph in Discussion session to speculate the possible clinical importance of our findings as follows:

“There have been many studies on modeling the prediction of function recovery after stroke. The representative models are the PREP algorithm and the proportional recovery model.[21] However, when the two models were validated, the predicted prognosis rate (sensitivity and specificity) remained around 73-88%. The reason is that fitting to that predictive model did not work well in patients with severe corticospinal tract damage early in the injury, clinically complete unilateral paralysis, and patients with no MEP response. That is, in the case of mild to moderate severity, the prediction through clinical data and infarction size fit well, but in the case of severe case, it was difficult to predict through the model, and these were called 'non-fitter'. In this study, we included relatively homogenous patients with MCA infarction, clinically no hand movement at all and neurophysiologically no MEP response. Therefore measuring interhemispheric functional connectivity in these patient populations can help predict prognosis.”

8) 22 stroke patients doesn't seem that large population. Authors should consider to mention this in the limitations.

(Author response) We completely agree with the reviewer on that it should be mentioned that our study is also limited by small numbers. Therefore, we added it as study limitation as follows:

“The present study had some limitations. First of all, our study is limited by small numbers in patient population. Due to this limitation, stroke patients included only those with cortical and subcortical stroke, and we could not definitely determine the differential impact of lesion location and stroke severity on rsFC.”

9) Line 299: this sentence is misunderstanding

(Author response) We deleted the sentence to avoid misunderstanding.

10) Lines 175-186: somehow journal instructions on how to write the paper got into this section

(Author response) We deleted line 175-186. This is where the journal's publication instruction was accidentally inserted. Thank you for your kind comment.

Round 2

Reviewer 2 Report

Dear authors,

after deep review of the revised manuscript, the authors improved the manuscript according to the two reviewer suggestions substantially. One of my main concerns, the potential conflict of deriving the seed coordinates for M1 and SMA based on task-fMRI could be ruled out by the authors' explanation and showing of the closeness of the SPM- and their result-driven coordinates for the ROIS.

Other minor things were changed and improved, according to our suggestions.

However, regarding the lesion volumes, one last major concern remains in my eyes: the authors plot the lesion overlap images now and give us significant differences about the lesion volumes between good and poor recovery group. This finding in their patient sample cannot simply be stated as additional sentence. This should be covered and included in their analysis model as a covariate. The authors should the lesion volume include in their statistical model and check to which amount the recovery is explained by the lesion volume and to which extent by their FC-connectivity values. I guess that the lesion volume alone will "swallow" a large amount of explanatory variance of the outcome parameter. Readers should know this.

Beyond this, I ask the authors to give an additional column in their table: I would like to see both real "total lesion volume" and "overlap volume" for each patient, not only the overlap volume.

Taken together, the study is still important and should enrich our knowledge on outcome prediction in stroke patients.

Author Response

We thank the reviewer for his/her attention and thoughtful comments. Below we have provided responses to specific comments, and noted changes in the manuscript.

Reviewer 2

Dear authors,

After deep review of the revised manuscript, the authors improved the manuscript according to the two reviewer suggestions substantially. One of my main concerns, the potential conflict of deriving the seed coordinates for M1 and SMA based on task-fMRI could be ruled out by the authors' explanation and showing of the closeness of the SPM- and their result-driven coordinates for the ROIS.

Other minor things were changed and improved, according to our suggestions.

1)However, regarding the lesion volumes, one last major concern remains in my eyes: the authors plot the lesion overlap images now and give us significant differences about the lesion volumes between good and poor recovery group. This finding in their patient sample cannot simply be stated as additional sentence. This should be covered and included in their analysis model as a covariate. The authors should the lesion volume include in their statistical model and check to which amount the recovery is explained by the lesion volume and to which extent by their FC-connectivity values. I guess that the lesion volume alone will "swallow" a large amount of explanatory variance of the outcome parameter. Readers should know this.

(Author response) We thank the reviewer for important and thoughtful comments. Following the suggestion by the reviewer, we did additional analysis, which included the total lesion volume as a covariate. The result was added in Results session as follows:

“When the total lesion volume was included as a covariate in statistical analysis, ANOVA F-test results revealed a statistically significant difference in FC between ipsilesional M1–contralesional M1 among the three groups (p=0.018) and in FC between ipsilesional M1–contralesional SMA among the three groups (p=0.015). In the post-hoc two-sample t-test, however, there was no difference in the FC value between the good recovery group and poor recovery group (p=0.232).”

We also add a sentence in Discussion session to describe that the lesion volume will "swallow" a large amount of explanatory variance of the outcome parameter as follows:

“However, the total lesion volume along showed a tendency to swallow a large amount of explanatory variance of the outcome parameter.”

2)Beyond this, I ask the authors to give an additional column in their table: I would like to see both real "total lesion volume" and "overlap volume" for each patient, not only the overlap volume.

(Author response) We revised Table 1 to include both real "total lesion volume" and "overlap volume" for each patient.

3)Taken together, the study is still important and should enrich our knowledge on outcome prediction in stroke patients.

(Author response) We deeply appreciate the reviewers’ high evaluation and encouragement on the importance of our study.
